# Analysis of a Three-Phase Induction Motor with a Double–Triple-Layer Stator Winding Configuration Operating with Broken Rotor Bar Faults

**Mbika Muteba** 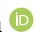

Department of Electrical and Electronic Engineering Technology, University of Johannesburg, Johannesburg 2092, South Africa; mmuteba@uj.ac.za

**Abstract:** This paper presents the performance analysis of a three-phase squirrel cage induction motor (SCIM) with a double–triple-layer (DTL) stator winding configuration operating with broken rotor bar (BRB) faults. The effects of BRB faults on the performance of specific parameters are analyzed under a steady-state regime. The SCIM is modeled using the two-dimensional finite element method (FEM) to study electromagnetic performance under healthy and BRB faulty conditions. To validate the finite element analysis (FEA) results, a prototype of an SCIM with a DTL stator winding configuration is tested for performance evaluation under healthy and BRB faulty conditions. The FEA and experimental (EXP) results of the SCIM with a DTL stator winding arrangement are compared with the results of the SCIM with a conventional double-layer (CDL) stator winding configuration. FEA and EXP results evidenced that the SCIM with a DTL stator winding configuration mitigates some of the adverse effects introduced by the BRB faults compared to the SCIM with a CDL stator winding of the exact specifications. Under loaded conditions, the SCIM with a DTL stator winding configuration reduced the magnitudes of the twice slip frequency sidebands caused by BRB faults from $\pm 1.2$ Hz for the SCIM with a CDL stator winding arrangement down to $\pm 0.2$ Hz and $\pm 0.36$ Hz when operating with 3BRB and 6BRB faults, respectively. The results also indicate that the SCIM with a DTL stator winding configuration has reduced the decibel sideband magnitude by 7.5 dB and 8 dB for unloaded and loaded conditions, respectively. This premise has positioned the SCIM with a DTL stator winding configuration as a strong candidate in applications where BRB faults are frequent, and the motor may be required to continue operating with a BRB fault until scheduled maintenance is in effect.

**Keywords:** broken rotor bar fault; conventional double-layer winding; current signature; double–triple-layer winding; experimental measurement; finite element analysis; performance analysis; squirrel cage; three-phase induction motor

## 1. Introduction

Three-phase squirrel cage induction motors are the most employed electrical machines in modern industry applications [1,2]. They are robust, consist of rotor bars that run axially through the rotor laminations, and are joined at the ends by end rings to form a cylindrical cage. Rotor-related faults account for about 10% of induction motor failures [3,4]. Although broken rotor bars do not initially cause an SCIM to fail, there can be severe secondary effects [3], such as the asymmetrical operation of the rotor, causing unbalanced currents, torque pulsation, increased losses, and decreased average torque [5]. To avoid the operation of induction motors with broken rotor bars, researchers proposed several condition-monitoring techniques to extract the broken rotor bar features [6–13]. In [6], an analysis of residual current using wavelets was conducted through extraction and removal of the fundamental component of currents to detect BRBs in induction machines with transient operating speeds [6]. A comprehensive review of the techniques proposed in the recent literature for BRB fault detection and diagnosis in line-fed and inverter-fed motors is

presented in [9]. In the latter, the diagnosis challenges, and signatures of the BRB faults are discussed [9]. Elsewhere, a two-step method for BRB detection has been proposed in [12]. The first step is analyzing the three-axis vibration signals using fractal dimension theory. The second method uses a fuzzy logic system to automatically diagnose the BRB faults in transient and steady-state regimes [12]. The work reported in [13] proposed the detection of BRB faults in SCIMs through music and zero-sequence. In that work, the zero-sequence current (ZSC) is analyzed using the high-resolution spectral method of multiple signal classification [13].

In applications with chemical and explosive atmospheres, vibrations, limited space, and rough treatments, SCIMs may be required to continue operating with BRBs until scheduled maintenance is in effect. One would expect to minimize the adverse effects when the SCIM is operating with BRBs. In [14], a new Neural–Fuzzi Controller has been proposed to mitigate the speed ripple of a vector-controlled SCIM when operating with BRBs [14]. An Indirect Field-Oriented controller processing currents and angular velocity of the SCIM to force desired flux and speed profiles to a fault compensation unit has been proposed in [15]. In the latter, a fault compensation unit is designed to have global tracking of the desired references and semi-global tolerance when the SCIM is operating with possible BRB faults [15]. Although the minimization methods suggested in [14,15] have some technical merits, they employed control algorithms that are complex and may bring additional maintenance costs for SCIMs in the petrochemical industry, gas terminals, and refineries onshore and other platforms offshore where there is a substantial amount of smaller and medium-sized SCIMs for various services.

This paper analyzes a three-phase squirrel cage induction motor with a double–triple-layer stator winding configuration for healthy and broken rotor bars' faulty operations. The ability of the double–triple-layer stator winding configuration to minimize some of the adverse effects of broken rotor bars is analyzed. The organization of this article is as follows: Section 2 briefly reports on key related work. Materials and methodology are provided in Section 3. Results and discussion are reported in Section 4, and Section 5 gives a summary of this article and possible future work.

## 2. Related Work

The airgap magnetic flux distribution in SCIMs is influenced by different factors, including the stator and rotor magneto-motive force distributions, magnetic saturation in stator and rotor teeth, and back-iron cores [16]. The placement of coils in slots leads to a stepped-like waveform of the stator or rotor magneto-motive forces, which exhibit space harmonics [17]. The winding arrangements have an influence on phase-belt harmonics, which depend on the deployment of coils under different phase belts. The work reported in [17] shows that the double–triple winding configuration mitigated the phase-belt harmonics and improved the airgap magnetic flux distribution of an SCIM. In a three-phase SCIM, the first and second phase-belt field harmonics are the fifth and seventh harmonics, respectively. The presence of broken bars introduces high-frequency components in the machine currents spectrum in addition to the characteristic sidebands around the fundamental component [18]. These additional components are due to the interaction between rotor asymmetry and either the voltage harmonics, winding distribution, or rotor slots [18]. The work reported in [18] analyzed the components at frequencies near the fifth and seventh harmonics, produced by the interaction between the BRB faults and the harmonics of the spatial distribution of stator windings. A multiple-coupled circuit model with both arbitrary winding layouts has been proposed in [19]. This model considered the actual distribution of stator windings and rotor bars and facilitated the calculation of the inductance for different fault conditions [19].

## 3. Materials and Methodology

### 3.1. Specifications and Ratings

The specifications and rating given in Table 1 are for SCIMs with conventional double-layer (CDL) and double–triple-layer (DTL) winding configurations. Both windings have been designed with the same number of turns per phase and the same conductor sizes. The SCIM stator frame has 36 slots, and the rotor has 43 bars. Both CDL and DTL winding configurations are designed to provide four magnetic poles, and the coils are chorded by one slot pitch.

**Table 1.** Specifications and ratings of the three-phase SCIM.

| Description | Values | Unit |
|---|---|---|
| Rated current | 12.6 | A |
| Rated power | 5.5 | kW |
| Nominal voltage | 380 | V |
| Nominal frequency | 50 | Hz |
| Rated speed | 1478 | rpm |
| Number of poles | 4 | - |
| Number of stator slots | 36 | - |
| Number of rotor bars | 43 | - |
| Number of turns per phase | 54 | - |
| External diameter | 210 | mm |
| Airgap length | 0.35 | mm |
| Core length | 160 | mm |

### 3.2. Double–Triple-Layer Stator Winding Configuration

The stator coils are laid in 36 slots to provide 4 magnetic poles rotating at synchronous speed $n_s$ = 1500 rpm. For CDL winding, there are three slots per pole per phase, constituting a phase belt of $\pi/3$ electric radian. Figure 1a,b show how coils are arranged in stator slots of the CDL and DTL winding configurations, respectively. The DTL winding configuration shown in Figure 1b has in some slots a mix of one full coil and two half coils belonging to two different phases. The currents in these phases are out of phase with one another by either $2\pi/3$ electric radians or $4\pi/3$ electric radians. To that effect, the net current and leakage flux are less in slots with currents belonging to different phases compared with slots with currents belonging to the same phase. The phase belt of the DTL winding configuration has been expanded from $\pi/3$ electric radians to $4\pi/9$ electric radians. This is because the coils of a phase in the DTL winding configuration occupy four slots under each pole for either the top or bottom layer. It is also notable in Figure 1b that the four slots of a phase band accommodate conductors that carry currents belonging to all three phases. Figure 2 shows the complete winding arrangement of the DTL winding configuration, and Table 2 compares the harmonic winding factors of both configurations.

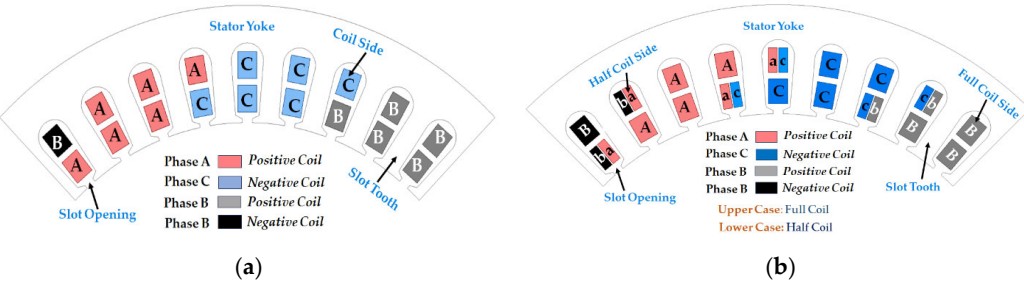

**Figure 1.** Winding arrangement in stator slots. Only one of the four poles in the stator is shown: (**a**) CDL winding configuration, (**b**) DTL winding configuration.

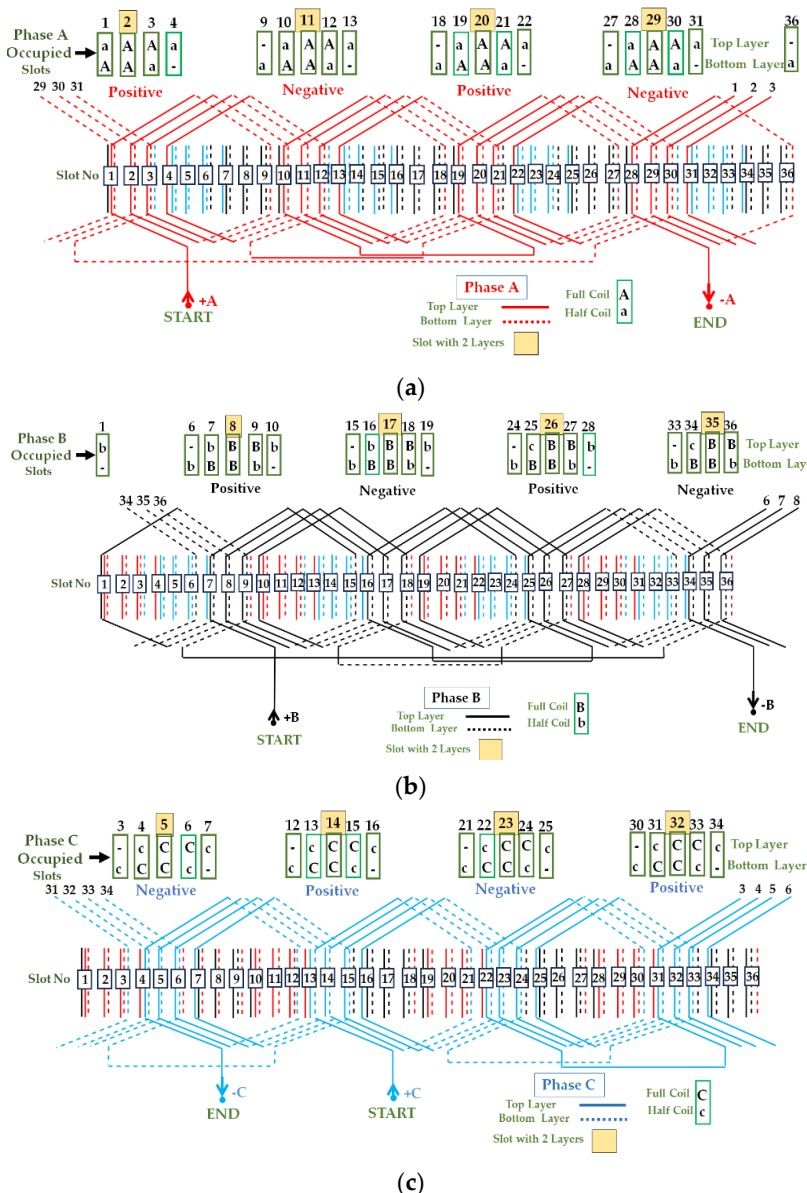

**Figure 2.** DTL winding configuration of the three-phase squirrel cage induction motor: (**a**) phase A, (**b**) phase B, (**c**) phase C.

**Table 2.** Comparison of key harmonic winding factors.

| | Harmonic Order | | | | | | |
|---|---|---|---|---|---|---|---|
| **Winding** | **1st** | **5th** | **7th** | **11th** | **13th** | **17th** | **19th** |
| CDL | 0.94 | 0.13 | −0.06 | −0.10 | −0.13 | −0.94 | −0.94 |
| DTL | 0.91 | −0.07 | −0.08 | −0.08 | −0.07 | −0.91 | 0.91 |

The general expressions for harmonic distribution factors $k_{dn}$ for the conventional double-layer winding and the proposed double–triple-layer are given by (1) and (2), respectively. The four slots per pole per phase that comprise the phase belt in the proposed DTL winding configuration have coils with different field component magnitudes. Two slots housed a half coil each, and the other two slots housed a full coil each. The harmonic "v"

distribution factor of the proposed DTL configuration is then calculated by also taking into consideration the difference in the slot field's component.

$$k_{d\ v\ (CDL)} = \frac{\sin v\ d\ (\beta_s/2)}{d\sin v(\beta_s/2)} \tag{1}$$

$$K_{d\ v\ (DTL)} = \frac{\sin v(d_i + d_j)(\beta_s/2)}{(d_i + d_j)\sin v(\beta_s/2)} \tag{2}$$

where $d_j$ is the number of slots per phase per pole that accommodate half coil sides and $d_i$ is the number of slots per pole per phase that accommodate full coil sides only, v is the harmonic order number, and $\beta_s$ is the slot pitch angle in electric degree. In Figure 1a, it is clearly notable that the phase coil sides of the CDL winding configuration extend into the adjacent pole by $1\beta_s$, whereas the phase coil sides of the proposed DTL winding configuration in Figure 1b extend into the adjacent pole by $2\beta_s$. The harmonic "v" pitch factor expressions of the CDL and proposed DTL winding configurations are, respectively, given by (3) and (4).

$$K_{p\ v(CDL)} = \cos\frac{1}{2}v(\tau_p - \gamma_c) \tag{3}$$

$$K_{p\ v\ (DTL)} = \cos\frac{1}{2}v[\tau_p - (\gamma_c - 2\beta_s)] \tag{4}$$

where $\gamma_c$ is the coil pitch angle of the CDL in an electrical degree and $\tau_p$ is the pole pitch angle in an electrical degree. In addition to the fundamental component, the stator currents will produce stator slot and winding phase-belt magnetomotive force (MMF) harmonics, respectively, given by (5) and (6). The rotor current will cause MMF slotting harmonics that have orders expressed by (7).

$$v_{ss} = x\frac{N_s}{p_1} + 1 \tag{5}$$

$$v_{sb} = x\ m + 1 \tag{6}$$

$$v_{rs} = x\frac{N_r}{p_1} + 1 \tag{7}$$

Here, $N_s$ is the number of stator slots, $p_1$ is the fundamental number of pole pairs, $N_r$ is the number of rotor slots, m is the number of phases, and x is any positive or negative integer number but not zero. In (5) and (6), the stator MMF harmonics have orders like $v_{ss}$ = −17, +19, −35, +37, −53, +55, −71, +73. . . and $v_{sb}$ = −5, +7, −11, +13, −17, +19, −23, +25, −29, +31. . .. . The 43 rotor cage bars are placed in slots. The stator MMF harmonics have orders like $v_{rs}$ = −21, +23, −43, +45, −65, +67. . .. .

### 3.3. Approach

The overview of the approach followed in this work is shown in Figure 3. In this work, the SCIMs are first designed and modeled using the finite element method. A two-dimensional (2D) FEA is performed using the ANSYS 22.0 electromagnetic package for magnetic static and AC magnetic transient solutions. The three-phase windings are excited by three-phase sinusoidal voltages, with zero initial current. Skin effect and core loss at 50 Hz are considered in the AC magnetic transient FEA. The broken rotor bar fault is achieved by modeling a vacuum in the specific rotor slot by completely removing the rotor copper bar. Secondly, the prototype SCIM is tested to validate the FEA results. the healthy squirrel cage rotor and the rotors with broken bars are shown in Figure 4. The experimental setup is shown in Figure 5. The broken rotor bar faults are achieved by drilling deep holes into the specific bars. The experimental setup comprises the tested three-phase SCIM coupled to a 7.5 kW induction motor fed from a Siemens SINAMICS four-quadrant energy recovery AC drive.

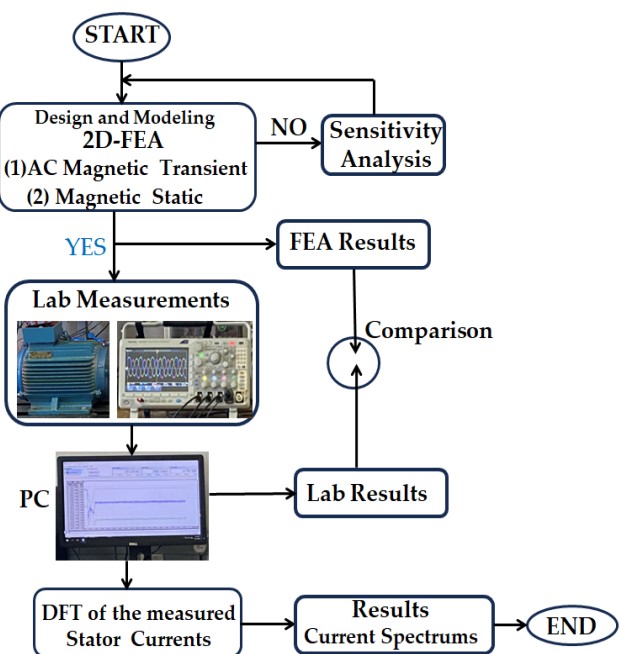

**Figure 3.** Overview of the approach.

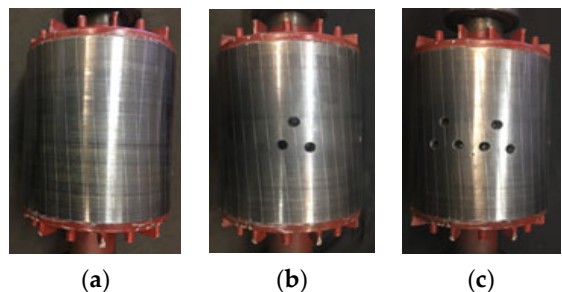

| **(a)** | **(b)** | **(c)** |

**Figure 4.** Squirrel cage rotor: (**a**) healthy rotor, (**b**) rotor with three broken rotor bars, (**c**) rotor with six broken rotor bars.

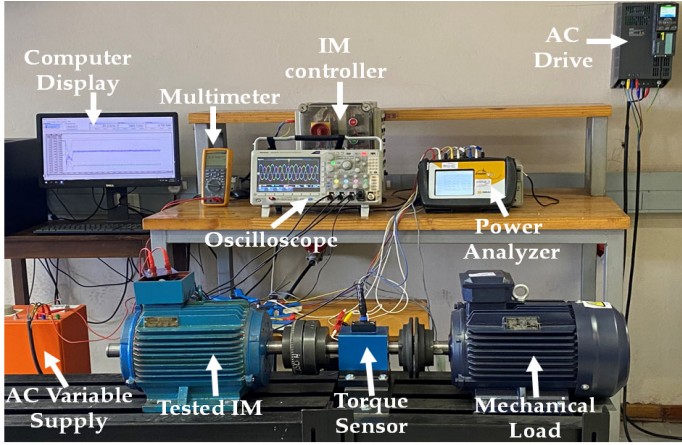

**Figure 5.** Experimental setup rig photo.

The 7.5 kW induction motor is used as a mechanical load. The shaft power, speed, and torque are measured by a DR-3000-P type torque transducer. A Dranetz PowerVisa 4400 three-phase power analyzer was used to measure the voltage, current, input power, and power factor, and a Tektronix MDO3000 Series digital oscilloscope was employed to store and analyze the current signatures. The direct-on-line start was made possible by start

and stop buttons incorporated into a motor controller. A variable AC power supply was used to vary motor voltage during no-load and locked rotor tests. The measured current waveforms are stored in a computer for further analysis. Lastly, the frequency features of the periodical signal of the currents were extracted by transforming the signal into a frequency domain using Discrete Fourier Transform (DFT).

The experimental (EXP) equivalent circuit parameters were obtained from the conventional no-load and locked rotor tests. The magnetizing reactance $X_m$ and the core loss resistance $R_c$ of the motors were obtained at a rated frequency of 50 Hz and a rated voltage of 380 V with the mechanical load removed from the shaft. On no-load, the SCIMs rotated at a speed close to 1500 rpm, which is the synchronous speed. The induced rotor currents were assumed to be very small, as the speeds of the slips were also very small. With approximately very small, induced currents, the stator currents on no-load were approximately equal to the magnetizing currents. The rotor resistances $R_r^s$ and total leakage reactance $X_l^s$ referred to in the stator were obtained from the locked rotor tests at a rated current of 12.6 A and a rated frequency of 50 Hz. On the other hand, the finite element analysis (FEA) equivalent parameters were obtained from the ideal no-load and locked rotor simulations. The ideal no-load simulation was used to determine the FEA magnetizing reactance through a magnetostatic solver. The current was assumed to be zero since the main objective of the no-load simulation by FEA is to compute the magnetizing inductance and to observe the magnetic saturation level in different parts of the SCIM [16,20]. The FEA total leakage reactance and rotor resistance referred to in the stator were obtained through locked rotor simulation. The AC magnetic transient solver was employed at zero speed, and the simulation was carried out at a rated current. The reason for keeping the current to its rated value stems from the fact that the leakage reactance is significantly affected by magnetic saturation. The speed has been zero; the output mechanical power is zero. When neglecting the iron losses during locked rotor simulation, the rotor input power is equivalent to the total copper loss in the stator winding and rotor bars, and it is directly obtained from FEA through an AC magnetic transient solver [20].

## 4. Results and Discussions

### 4.1. Effect of Broken Rotor Bars on Flux Density

The effect of the number of broken rotor bars on a 3 kW, four pole, thirty-six stator slot, forty-four rotor bar three-phase cage induction motor under different loading and speed conditions is analyzed in [21]. The results in [21] evidenced a relative increase in the magnitude of variance for the growing number of broken bars regardless of the operating point. There is some uncertainty when seeing the difference between a healthy and one broken bar at lower loads [21]. As a result, only two high degrees (three and six broken bars) of rotor bar faults are analyzed in this paper to clearly elaborate the difference in performance between the CDL and DTL squirrel cage induction motors. Figure 6a–c show the no-load flux density distribution in the iron cores of the SCIM with a CDL winding configuration under healthy, faulty with 3BRB, and faulty with 6BRB, respectively. On the other hand, Figure 7a–c show the no-load flux density distribution in the iron cores of the SCIM with a DTL winding configuration under healthy, faulty with 3BRB, and faulty with 6BRB, respectively. The flux density distributions in Figures 6 and 7 lie in the cartesian plane of the model. The vector boundary condition with zero vector potential is set to the outer region of the machines' models. The broken rotor bars in the SCIM are considered a type of asymmetric operation of the motor's rotor circuit [9]. The motor rotor's circuital asymmetry results in an increase in the backward magnetic field. The latter leads to the deformation of the motor magnetic flux density spatial distribution by creating undesirable highly saturated regions around the broken bars, as noted in Figure 6b,c and Figure 7b,c. The airgap flux density profiles of the unloaded SCIMs are shown in Figures 8–10.

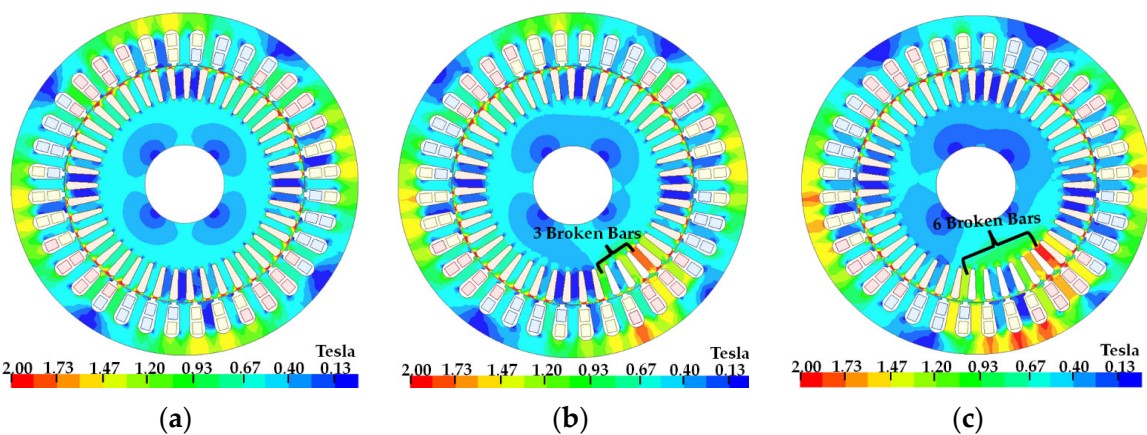

**Figure 6.** Flux density distribution on the full load of the IM with the CDL: (**a**) healthy, (**b**) 3BRB, (**c**) 6BRB.

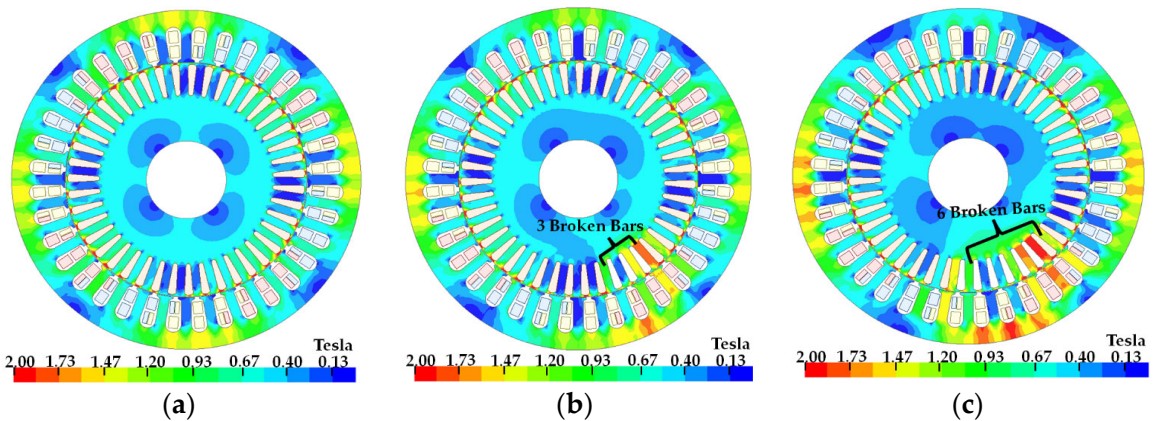

**Figure 7.** Flux density distribution on the full load of the IM with the DTL: (**a**) healthy, (**b**) 3BRB, (**c**) 6BRB.

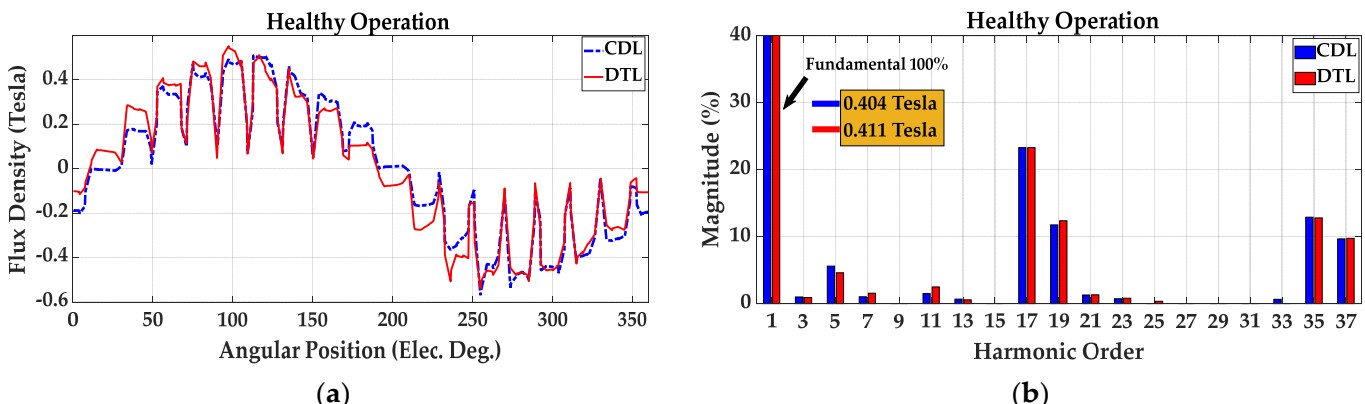

**Figure 8.** Airgap flux density characteristic of the healthy machines under no-load operation: (**a**) airgap flux density profile, (**b**) FFT of the airgap flux density profile.

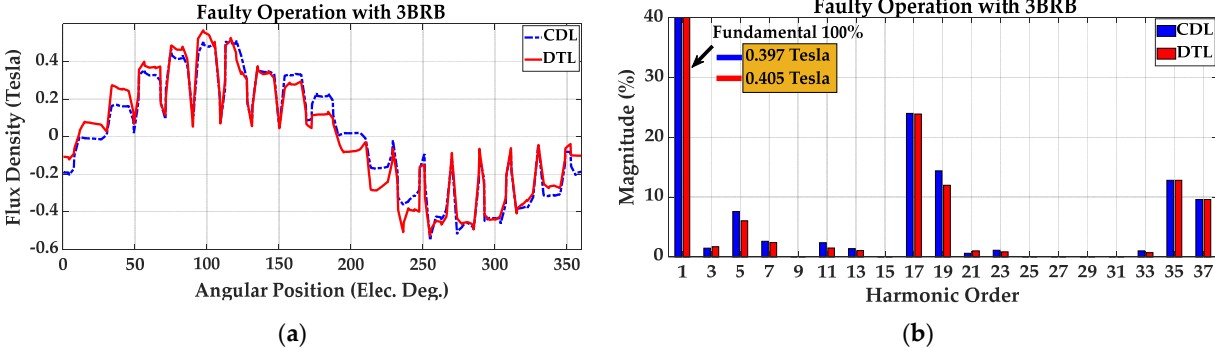

**Figure 9.** Airgap flux density characteristic of the faulty machines with 3BRB under no-load operation: (**a**) airgap flux density profile, (**b**) FFT of the airgap flux density profile.

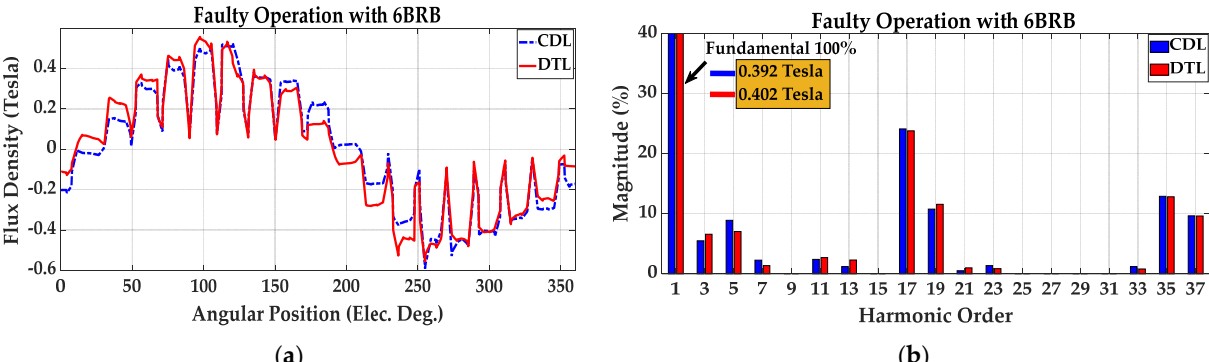

**Figure 10.** Airgap flux density characteristic of the faulty machines with 6BRB under no-load operation: (**a**) airgap flux density profile, (**b**) FFT of the airgap flux density profile.

The magnetizing component of the no-load current given in Equation (8) is responsible for setting up the airgap magnetic flux in the SCIM, leading to a better overloading capability of the motor. To better understand the impact of the broken rotor bar on the airgap flux density, the SCIMs were first modeled and simulated for the unloaded current operation.

$$I_m = \frac{\pi V_s}{12\mu_o f_{s1}\tau_p L_e} \frac{p_1 g k_z (1 + k_F)}{(T_s k_{w1})^2} \tag{8}$$

In Equation (8), $k_{w1}$ denotes the fundamental winding factor, $p_1$ indicates the fundamental number of pole pairs, $f_{s1}$ indicates the fundamental frequency of the current, $T_s$ denotes the stator number of turns in series per phase, $k_z$ denotes Carter's coefficient, $k_F$ indicates the saturation factor, $V_s$ is the stator phase voltage, $L_e$ denotes the effective stator stack length, $\tau_p$ denotes the pole pitch, g indicates the airgap length, and $\mu_s$ denotes the magnetic permeability of the vacuum. The radial component of the airgap flux density is given in Equation (9), and it is dependent on the permeance of the airgap and the magnetomotive forces (MMFs) of the stator and rotor currents.

$$B_g(\theta_m, t) = \lambda(\theta_m, t)[M_s(\theta_m, t) + M_r(\theta_m, t)] \tag{9}$$

where $\lambda$ is the permeance of airgap per area, $\theta_m$ is the mechanical angle, and $M_s$ and $M_r$ are the magnetomotive forces (MMFs) of the stator and rotor current, respectively. The magnetic permeance of the airgap given by:

$$\lambda(\theta_m, t) = \frac{\mu_o}{g k_z} + \sum_{a=1}^{\infty} \lambda_{s,\,a} \cos(a Z_s \theta_m) + \sum_{b=1}^{\infty} \lambda_{r,\,b} \cos[a Z_r(\theta_m - t)] + \xi_{a,\,b}(\theta_m, t) \tag{10}$$

$$\xi_{a,\,b}(\theta_m, t) = \sum_a^\infty \sum_b^\infty \lambda_{s,\,r\,(a\pm b)} \cos[(aZ_s \pm b\,Z_r)\theta_m - bZ_r\omega_r t] \tag{11}$$

Here, a and b denote the integers associated with the stator and rotor slots, respectively, and $\lambda_s$ and $\lambda_r$ denote the partial magnetic permeances of the stator and rotor, respectively. Equation (10) suggests that the magnetic permeance is influenced by the stator and rotor slot openings. Carter's coefficient $k_z$ in Equation (10) accounts for both stator and rotor slot openings, which are clearly visible in the airgap flux density profiles shown in Figures 8–10. It is also noted in Equation (10) that magnetic permeance has spatial harmonics related directly to the number of stator and rotor slots $Z_s$ and $Z_r$, respectively. The MMFs of the stator and rotor currents are given by:

$$M_s(\theta_m, t) = \sum_{v=2mx\pm1}^\infty M_{s1}\cos(vp_1\theta_m \pm \omega_{s1}t) \tag{12}$$

$$M_r(\theta_m, t) = \sum_{v=1}^\infty M_{r1}\cos(v\theta_m \pm \omega_{s1}t - \varphi_o) \tag{13}$$

Here, $M_{s1}$ and $M_{r1}$ indicate the amplitude of the stator and rotor fundamental MMFs, respectively, $\omega_{s1}$ is the fundamental angular frequency of the stator current, and $\varphi_o$ is the reference angle between the stator and rotor fundamental MMFs. The stator fundamental MMF on no-load can be computed as:

$$M_{so1} = k\,I_o = \frac{3\sqrt{2}T_s k_{d1} k_{p1}}{\pi p_1}(I_c + jI_m) \tag{14}$$

In Equation (14), $I_c$ denotes the core loss component of the no-load current. Since $T_s$, $k_{d1}$, $k_{p1}$, and $p_1$ are fixed, the stator fundamental MMF only varies as a function of the no-load current, which is mostly dominated by its magnetizing component, $I_m$, expressed in Equation (8). On no-load, the slip is very small, the equivalent rotor resistance is relatively high, and the frequency of the rotor current is quite very small, which results in a tremendous decrease in the rotor magnetizing reactance, making the rotor current on no-load $I_{ro}^s$ very small and negligible. At rated load or slip, the rotor circuit is dominated by the resistance $R_r^s/s_n$, thus the magnetizing current $I_m$ lags the nominal rotor electric current $I_{rn}^s$ by 90° [16]. On no-load, the rotor magnetomotive force $M_r(\theta_m, t)$ is negligible and it does not affect the airgap flux density on no-load, which can be expressed by:

$$B_{go}(\theta_m, t) = \lambda(\theta_m, t)M_{so}(\theta_m, t) \tag{15}$$

Here, $M_{so}$ is the stator MMF of the unloaded machine. In $(R_r^s / s_n)$, $R_r^s$ indicates the rotor resistance referred to in the stator and $s_n$ is the rated or nominal slip. The no-load airgap flux density characteristics in Figures 8–10 are mainly influenced by the airgap magnetic permeance, which accounts for the stator and rotor slotting, the stator winding arrangement, and the magnitude of the magnetizing current. Both the CDL and DTL winding configurations produced the first winding phase belt (−5th) field harmonics, which rotate in opposing directions to the fundamental field at the speed of $n_s/5$. They both also produced the second winding phase (+7th) belt airgap field harmonics, which rotate in the same direction as the fundamental field at the speed of $n_s/7$. Under healthy operation, the DTL winding, compared with the CDL winding, reduced the no-load fifth airgap flux harmonics from 5.6% down to 4.5%. With 3BRB, the CDL winding increases the no-load fifth airgap flux harmonics by 2%, from 5.6% up to 7.6%, and the DTL winding increases its no-load fifth airgap flux harmonics by only 1.5%, from 4.5% to 6.0%, under the same broken rotor bar fault. With 6BRB, the CDL winding increases the no-load fifth airgap flux harmonics by 1.3%, from 7.6% to 8.9%, while the DTL winding increases the no-load fifth airgap flux harmonics by only 0.6%, from 6.0% to 6.6%. The first slot harmonics (−17th), which rotate in opposing directions to the fundamental field at the speed of $n_s/17$, are dominant in both winding configurations under different working conditions. The no-

load 17th airgap flux harmonics are not impacted by the faulty broker rotor bar. The third airgap flux harmonics increased in both CDL and DTL winding configurations. An increase in magnetic saturation in the stator back iron near the region on the axis with broken rotor bars is noticed in Figures 6 and 7. When the SCIMs are loaded, an increase in rotor current results in a secondary armature reaction, which induces additional currents of a frequency different from the frequency of the supply in the stator windings [16]. Neglecting the end ring resistance, the rotor fundamental MMF of the loaded healthy SCIM can be computed as:

$$M_{r1} = kI_b = \frac{N_b}{\pi p_1}\sqrt{2}\left[\frac{R_b(I_b/J_{bco})}{\rho_{co}}\right] \tag{16}$$

In Equation (16), $N_b$ denotes the number of rotor bars, $I_b$ denotes the RMS value of the bar current, $R_b$ denotes the bar resistance, $J_{bco}$ denotes the copper bar current density, and $\rho_{co}$ denotes the copper resistivity at 25 °C. The current in the rotor bar is not uniform, and it depends on the rotor frequency. The rotor bar resistance in Equation (16) can be expanded to (17), and the rotor fundamental MMF of the loaded healthy SCIM can be computed using Equation (18).

$$R_r = \rho_{co}\frac{L_b}{(I_b/J_{bco})}\frac{R_{ac}}{R_{dc}} \tag{17}$$

$$M_{r1} = \frac{R_{ac}}{R_{dc}}\frac{N_b}{\pi p_1}\sqrt{2}L_b = K_R\frac{R_{ac}}{R_{dc}} \tag{18}$$

In Equation (18), the constant $K_R = \left(L_b\frac{N_b}{\pi p_1}\sqrt{2}\right)$, and $L_b$ indicates the rotor bar length. The rotor fundamental MMF of the loaded SCIM mainly depends on the ratio between the bar AC and DC resistances, $R_{ac}$ and $R_{dc}$, respectively. The impact of the rotor fundamental MMF of the loaded SCIM is analyzed through an FEA of the airgap flux density, as shown in Figures 11 and 12. The airgap flux densities of the loaded SCIMs operating with 3BRB and 6BRB for both CDL and DTL winding configurations are observed to increase the backward fifth airgap flux harmonics. The 3BRB fault increased the backward fifth airgap flux harmonics from 7.3% to 17.29% for the CDL winding and from 4.8% to 14.8% for the DTL winding. The 6BRB fault increased the backward fifth airgap flux harmonics from 7.3% to 15.1% for the CDL winding and from 4.8% to 11.5% for the DTL winding. On the other hand, the 3BRB fault decreased the forward seventh airgap flux harmonics from 3.3% down to 0.8% for the CDL winding and from 2.6% down to 1.8% for the DTL winding. In contrast to the 3BRB fault, the 6BRB fault increased the forward seventh airgap flux harmonics from 3.3% to 5.8% for the CDL winding and from 2.6% up to 5.9% for the DTL winding. It is further observed that under faulty operations, the third airgap flux harmonics increased in both CDL and DTL winding configurations. An increase in the third airgap flux harmonic is significant when the SCIMs operate with a 6BRB fault. The high third airgap flux harmonics are due to magnetic saturation in the stator back iron near the region of the broken rotor bars.

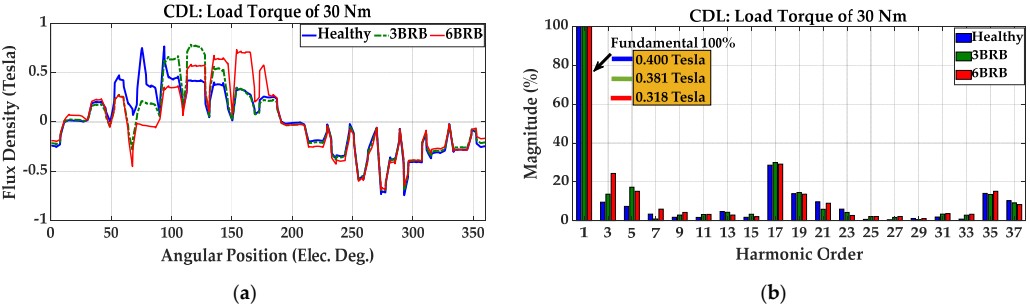

**Figure 11.** Airgap flux density characteristic of the SCIM with a CDL winding operating with load torque of 30 Nm: (**a**) airgap flux density profile, (**b**) FFT of the airgap flux density profile.

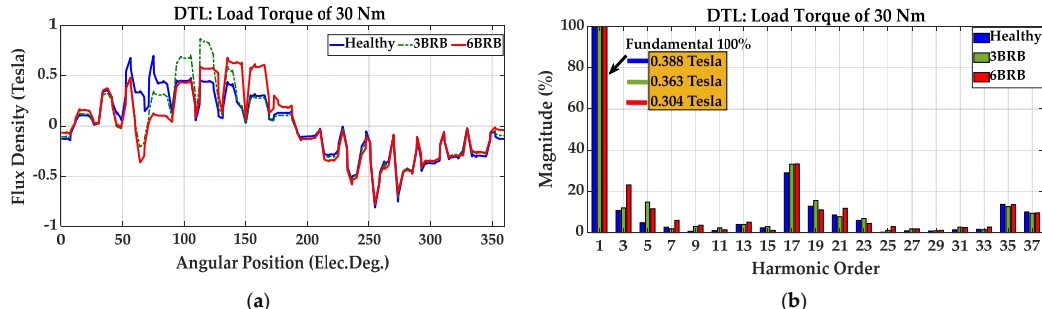

**Figure 12.** Airgap flux density characteristic of the SCIM with a DTL winding operating with load torque of 30 Nm: (**a**) airgap flux density profile, (**b**) FFT of the airgap flux density profile.

### 4.2. Effect of Broken Rotor Bars on the Current Signature of the Loaded SCIM

The operation of an SCIM with broken rotor bar faults introduces asymmetry in the rotor current distribution because there will not be a current flowing through the broken rotor bar. The latter introduces a disturbance in the airgap magnetic field distribution that induces voltages back in the stator winding, causing a current to flow through the winding and the supply. The forward and backward fundamental airgap rotating magnetic fields introduce currents at frequencies of $(1 - 2s)f_s$ and $(1 + 2s)f_s$, respectively. Here, s is the slip per unit and $f_s$ is the frequency of the supply.

Figures 13 and 14 show the FEA current characteristics of the loaded SCIMs. In both SCIMs with CDL and DTL stator winding configurations, the RMS values of the current increased when the motors operated with broken rotor bar faults. Furthermore, it is observed in Figures 13a–c and 14a–c that the stator current waveforms become more distorted when the SCIMs operate with broken rotor bar faults. The frequencies of the sidebands due to broken rotor bar faults are visibly noted in the FFT current spectrum shown in Figures 13d and 14d. Under healthy operation, the frequency of both left and right sidebands has a low amplitude and small slip compared with 3BRB and 6BRB faulty operations. Higher frequency components of the current spectrums are also notable in Figures 13d and 14d. The space harmonics due to the winding phase belt $f_{sb} = f_s(x\,m + 1)$ appear as zero in the phase current spectrums at exactly 250 Hz 350 Hz..., but with no-zero components in their surrounding frequencies, indicating the presence of broken rotor bar faults. Magnetic saturation causes the third harmonics to appear as no-zero components of the phase current spectrums around 150 Hz.

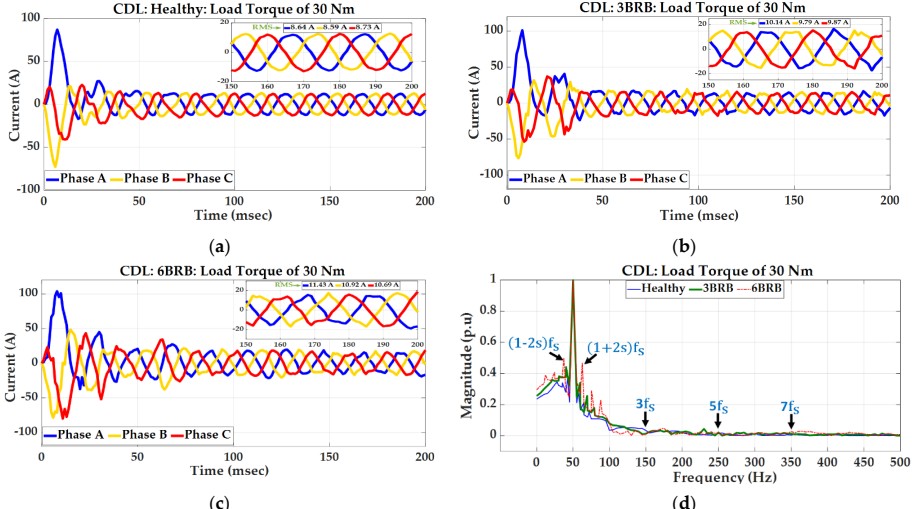

**Figure 13.** FEA loaded current characteristics of the SCIMs with a CDL winding configuration: (**a**) current waveforms for healthy operation, (**b**) current waveforms for faulty operation with 3BRB, (**c**) current waveforms for faulty operation with 6BRB, (**d**) FFT current spectrum of phase A recorded at 150 ms and 200 ms.

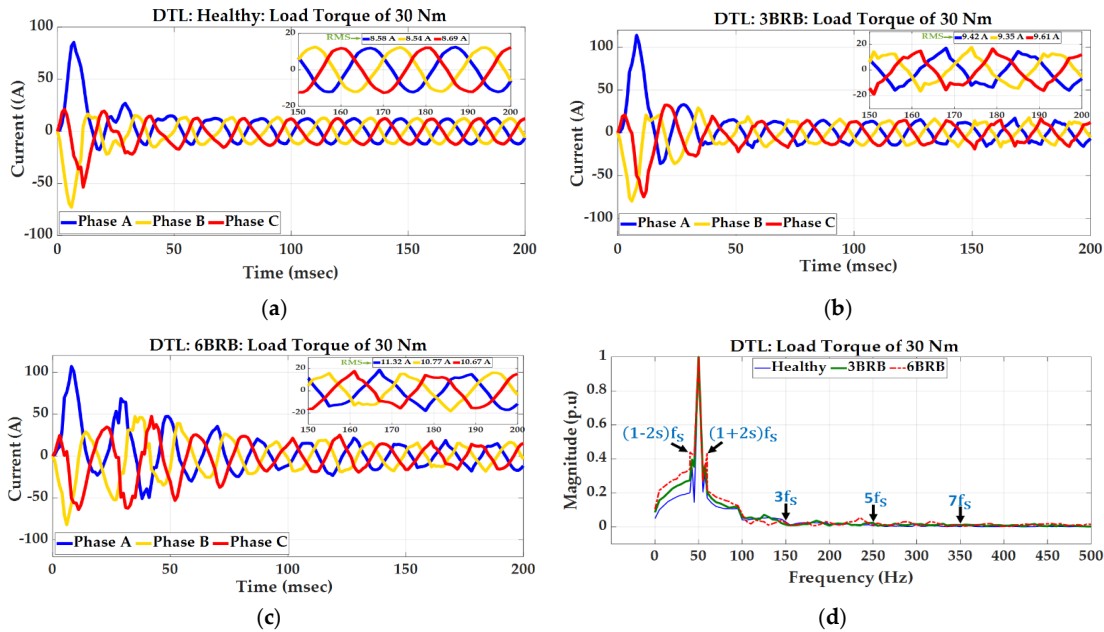

**Figure 14.** FEA loaded current characteristics of the SCIMs with a DTL winding configuration: (**a**) current waveforms for healthy operation, (**b**) current waveforms for faulty operation with 3BRB, (**c**) current waveforms for faulty operation with 6BRB, (**d**) FFT current spectrum of phase A recorded at 150 ms and 200 ms.

*4.3. Machines' Parameters*

The experimental (EXP) and FEA equivalent circuit parameters are given in Tables 3 and 4. Both EXP and FEA results in Tables 3 and 4 evidence that the magnetizing reactance decreases when the SCIM operates with broken rotor bars. A decrease in the magnetizing reactance results in an increase in the no-load current. Although the total leakage reactance referred to in the stator may be affected by magnetic saturation and the rotor frequency, which is dependent on the speed of slip per unit, the EXP and FEA results in both Tables 3 and 4 show no significant change when the SCIM operates with broken rotor bars. However, the rotor resistance referred to in the stator exhibits a decrease in operation with broken rotor bars. The change in rotor resistance referred to in the stator is caused by the increase in the speed of slip when the SCIM operates with broken rotor bars. When the rotor resistance referred to in the stator decreases, the rotor current referred to in the stator increases. The line stator current also increases, as observed in the results of measured currents shown in Figures 15 and 16.

**Table 3.** Parameters of the SCIM with a CDL.

| | Healthy | | 3BRB | | 6BRB | |
|---|---|---|---|---|---|---|
| Parameters | EXP | FEA | EXP | FEA | EXP | FEA |
| $X_m(\Omega)$ | 56.32 | 60.35 | 48.16 | 52.07 | 35.48 | 43.78 |
| $R_c(\Omega)$ | 220.59 | - | 213.54 | - | 188.06 | - |
| $X_l^s(\Omega)$ | 3.03 | 3.27 | 2.97 | 3.19 | 2.79 | 3.08 |
| $R_r^s(\Omega)$ | 1.12 | 0.97 | 1.03 | 0.87 | 0.94 | 0.81 |

**Table 4.** Parameters of the SCIM with a DTL.

| | Healthy | | 3BRB | | 6BRB | |
|---|---|---|---|---|---|---|
| Parameters | EXP | FEA | EXP | FEA | EXP | FEA |
| $X_m(\Omega)$ | 46.31 | 57.60 | 41.81 | 50.12 | 34.97 | 41.17 |
| $R_c(\Omega)$ | 207.28 | - | 191.54 | - | 189.53 | - |
| $X_l^s(\Omega)$ | 2.84 | 3.17 | 2.77 | 3.12 | 2.72 | 2.98 |
| $R_r^s(\Omega)$ | 1.24 | 0.88 | 0.97 | 0.81 | 0.90 | 0.75 |

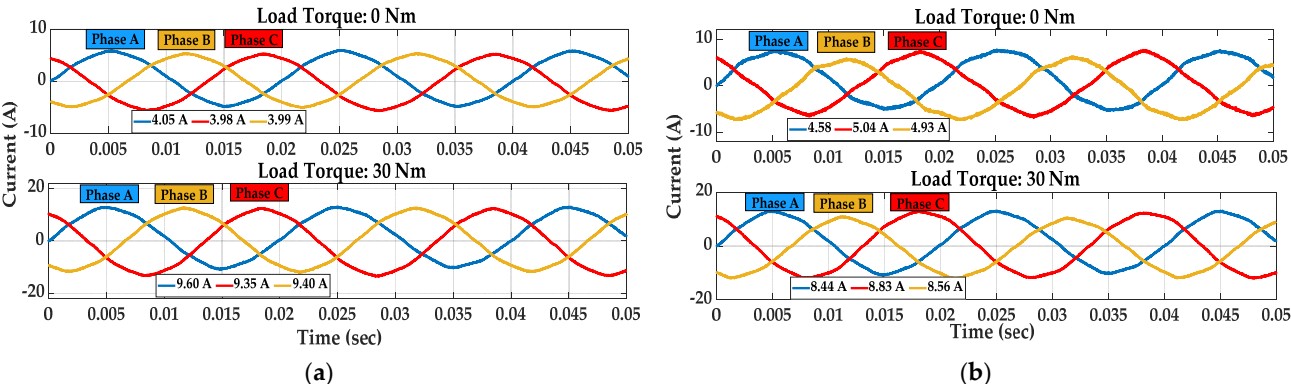

**Figure 15.** Steady-state stator currents under healthy operation: (**a**) CDL; (**b**) DTL.

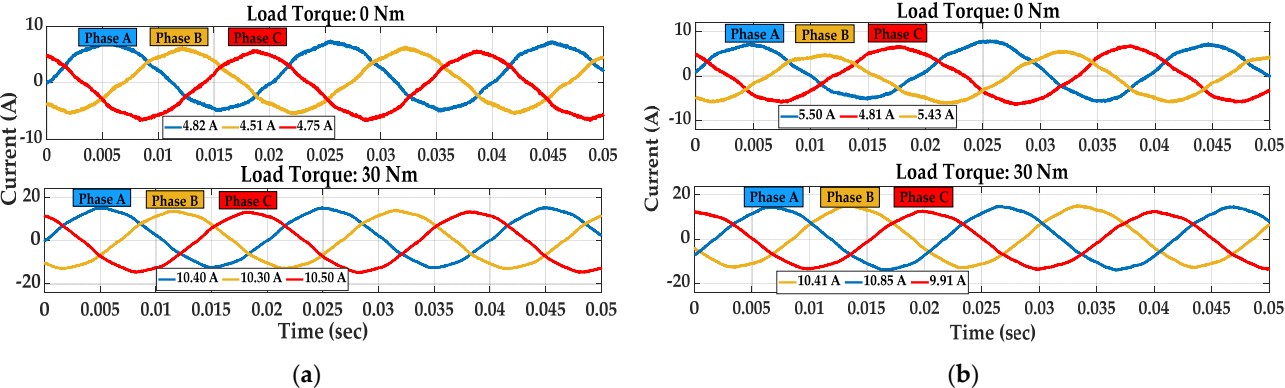

**Figure 16.** Steady-state stator currents under operation with 3BRB: (**a**) CDL; (**b**) DTL.

*4.4. Analysis of Measured Stator Currents*

Figures 15–17 show the measured steady-state current waveforms for unloaded and loaded SCIMs. Under healthy operation, the RMS values of the unloaded currents for the SCIM with CDL winding are less compared with the SCIM with a DTL operating under the same conditions. The reduced unloaded current for the SCIM with a CDL winding configuration is because its magnetizing reactance is high compared with the DTL winding configuration, as reported in the previous sub-section. The magnetizing reactance and the line stator current can be obtained using (19) and (20), respectively.

$$X_m = \frac{12\mu_o f_{s1}}{\pi}(N_s k_w)^2 \frac{\tau L_e}{p_1 g k_z (1+k_f)} \tag{19}$$

$$I_s = I_m + I_r^s = \frac{\pi V_s}{12\mu_o f s_1 \tau L_e} \frac{p_1 g k_z (1+k_F)}{(N_s k_w)^2} + I_r^s \tag{20}$$

The derivation of Equation (20) does not consider the core loss component of the current. It should be noted that the rotor current, referred to as the stator $I_r^s$, is much smaller than the magnetizing current during no-load motoring operations. The stator line current is mostly magnetizing, and it depends mainly on the supply voltage, winding configuration, and saturation factor.

The SCIMs have been connected in a star, at a rated supply voltage, and the slot teeth and back cores are saturated, as indicated in Section 4. The SCIMs with a DTL stator winding configuration exhibit more magnetic saturation than the SCIMs with a CDL winding configuration for healthy and broken rotor bar operations. Magnetic saturation introduces the third space harmonics, and their existence is notable in the FEA results shown previously in Figures 8–10.

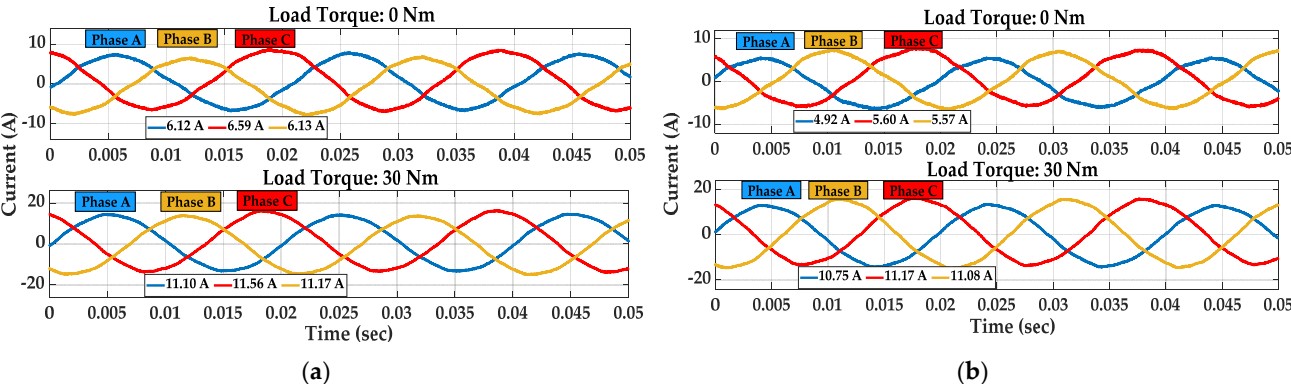

**Figure 17.** Steady-state stator currents under operation with 6BRB: (**a**) CDL; (**b**) DTL.

Observing the measured results in Figures 15–17, it is notable that the RMS value of unloaded line currents of the SCIMs with CDL winding configurations becomes greater compared to the RMS value of unloaded line currents of the SCIM with a DTL winding configuration when operating with 6BRB. Although the SCIMs have been connected in a star, the third harmonics dominantly occur in the FFT of the measured unloaded currents shown in Figures 18 and 19. One key observation must be made here. The third harmonics of the unloaded line currents of the SCIMs with CDL winding configurations decreased when operating with broken rotor bars. On the other hand, the third harmonics of the unloaded line currents of the SCIMs with a DTL winding configuration remain almost unchanged under healthy and broken rotor bar operations, maintaining the RMS values of the unloaded current that are almost constant under healthy and broken rotor bar operations.

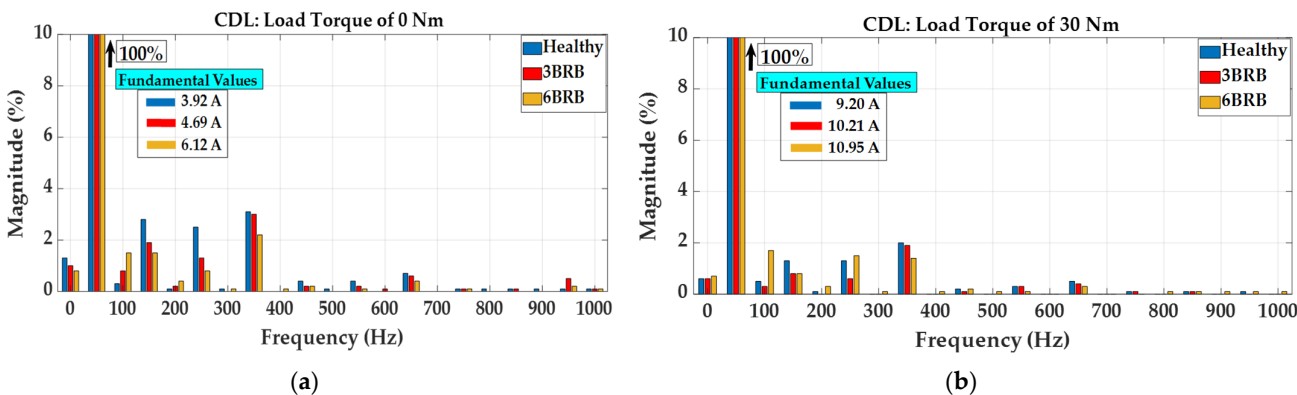

**Figure 18.** Harmonic components of phase A current profiles for the CDL: (**a**) operation with a load torque of 0 Nm, (**b**) operation with a load torque of 30 Nm.

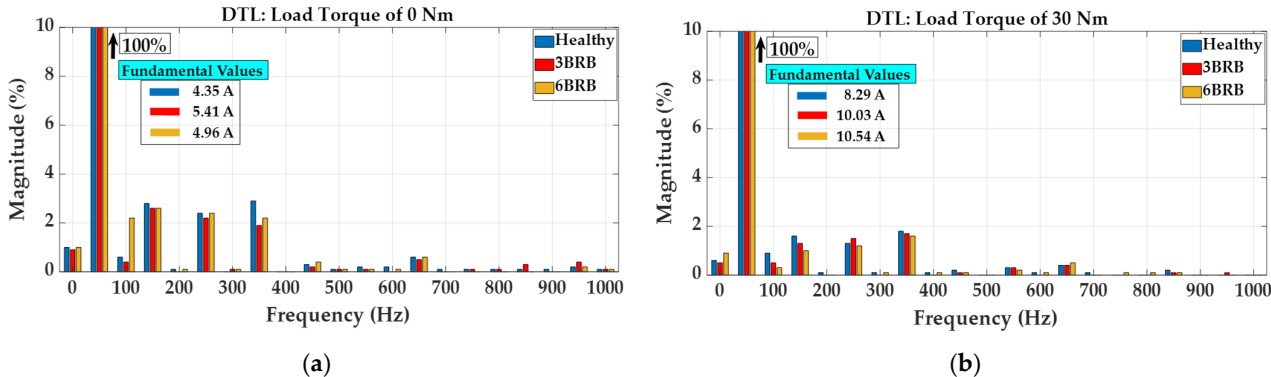

**Figure 19.** Harmonic components of phase A current profiles for the DTL: (**a**) operation with a load torque of 0 Nm, (**b**) operation with a load torque of 30 Nm.

Furthermore, it is noted that the RMS values of the measured loaded line currents of the SCIM with a CDL winding configuration are slightly higher than the loaded line currents of the SCIM with a DTL winding configuration when operating under healthy conditions. This is due to the high value of the fundamental component of the line current for the SCIM with a CDL winding configuration, as shown in Figure 18b. Under broken rotor bar faulty operations, the RMS values of the measured loaded line currents of the SCIM with a CDL winding configuration are almost the same as the loaded line currents of the SCIM with a DTL winding configuration. The frequencies of the current components in Figures 18 and 19 are indicative of the influence of an unbalanced rotor flux caused by the broken rotor bars, which has influenced the airgap field distribution, as noticed in FEA results. Furthermore, the unbalanced rotor flux must be considered as the combination of positive and negative sequence rotor fluxes rotating at the slip frequency, and the current harmonics can be observed as twice the slip of the frequency beside the fundamental frequency, as shown in the DFT zoom-measured current spectrums shown in Figures 20 and 21.

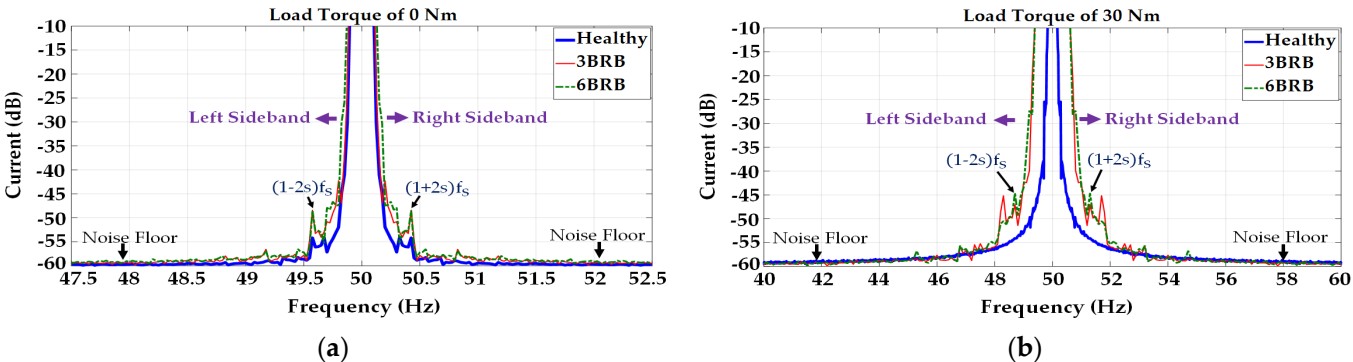

**Figure 20.** The DFT zoom current spectrum of phase A under the steady state of the SCIM with a CDL winding configuration: (**a**) operation with a load torque of 0 Nm, (**b**) operation with a load torque of 30 Nm.

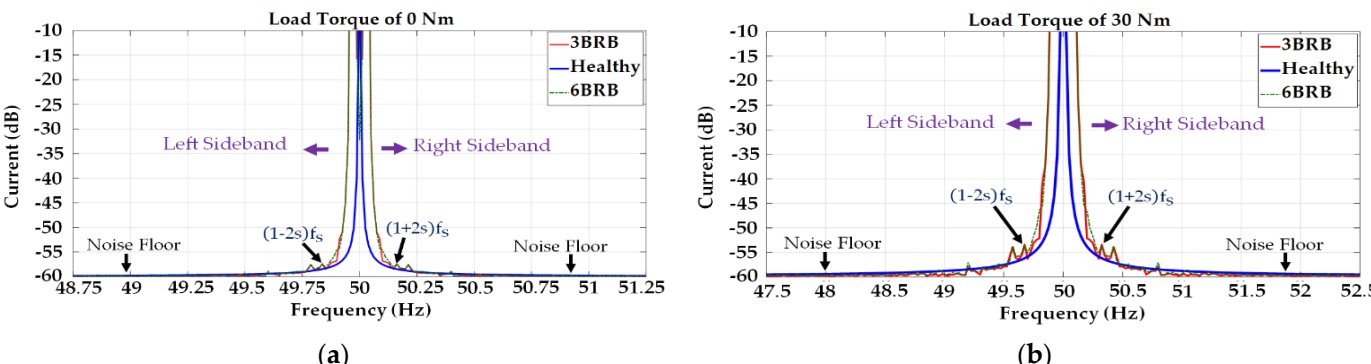

**Figure 21.** The zoomed DFT current spectrum of phase A under the steady state of the SCIM with a DTL winding configuration: (**a**) operation with a load torque of 0 Nm, (**b**) operation with a load torque of 30 Nm.

The magnitudes of the twice slip frequency sidebands $(1 \pm 2s)f_s$ due to broken rotor bars are clearly pointed out in the DFT zoom-measured current spectrums. The slip depends on the operation speed and motor load. As the mechanical load increases, the characteristic frequency of the BRB fault moves away from the fundamental frequency [22]. The slip frequency of the unloaded SCIM is minimal, and the twice slip frequencies under BRB faulty operation are close to the value of the supply frequency. This makes the detection of BRB through the analysis of current signatures less dependable when the SCIM is operating under light load conditions [23].

Observing Figure 20a, it is notable that the unloaded SCIM with the CDL stator winding configuration exhibits the twice slip sidebands at ±0.4 Hz around the supply frequency when operating either with 3BRB or 6BRB. In Figure 20b, it is clearly notable that the loaded SCIM with the CDL stator winding configuration exhibits the twice slip sidebands at ±1.2 Hz around the supply frequency when operating either with 3BRB or 6BRB. On the other hand, it is observed in Figure 21, under the same BRB faulty conditions that the SCIM with the DTL stator winding configuration exhibits the twice slip sidebands at ±0.2 Hz and ±0.36 Hz around the supply frequency for unloaded and loaded currents, respectively. Observing Figure 20a, the sidebands are about 40 dB down on the supply unloaded current of the SCIM with the CDL stator winding configuration for both 3BRB and 6BRB faulty operations. In Figure 20b, the decibel difference between the sideband magnitudes and the supply frequency components for the loaded current of the SCIM with the CDL winding configuration is about 35 dB. The magnitude of sidebands has increased by 5 dB from the unloaded current to the loaded current. On the other hand, the sidebands of the SCIM with the DTL winding configuration for both 3BRB and 6BRB faulty operations are about 47.5 dB and 43 dB down on the supply unloaded current in Figure 21a and loaded current in Figure 21b, respectively. The magnitude of sidebands has increased by 4.5 dB from the unloaded current to the loaded current. The measured results in Figures 20 and 21 clearly indicate that the SCIM with the DTL winding configuration compared with the SCIM with the CDL winding configuration decreased the magnitude of sidebands by 7.5 dB and 8 dB for unloaded and loaded currents, respectively. This is an indication that the SCIM with the DTL winding configuration can reduce the severity of broken rotor bar faults when operating with unloaded and loaded line currents.

*4.5. Analysis of Key Performance Parameters*

In this sub-section, key performance parameters of the SCIM with the DTL winding configuration are compared with those of the SCIM with the CDL winding configuration. Figure 22a–c show the steady-state measured shaft torque of the loaded motors. Tables 5 and 6 provide a comparison between the experimental (EXP) and finite element analysis (FEA) results of key performance parameters. Observing Figure 22a, the SCIM with the DTL winding configuration maintained the average torque and reduced the torque ripple by 16.2% when operating under healthy conditions. Figure 22b,c evidence that the torque ripple increases, and the average torque decreases when the SCIMs are operating with broken rotor bar faults. The torque ripples increase by 251% and 160% for the SCIM with the CDL winding configuration and the SCIM with the DTL winding configuration, respectively, when operating with a 3BRB fault. Under 6BRB faulty operations, the torque ripples increase by 312% and 208% for the SCIM with the CDL winding configuration and the SCIM with the DTL winding configuration, respectively.

The values of efficiency in Tables 5 and 6 were indirectly determined according to the IEC 60034-2-1 standard [24,25]. The additional load losses $P_{ALL}$ were measured starting from the residual losses [26,27], which were determined for each load point by subtracting the output power, the total iron losses $P_{Ti}$, the rotational losses, and the total copper losses $P_{TCu}$ from the input power. The process of determining all these losses is detailed in [26–30]. The efficiency decreased when the motors operated with BRB faults. This is due to the increase in the machines' total copper losses $P_{TCu}$ and total iron losses $P_{Ti}$ when operating with BRB faults. The increase in the total iron losses is due to a decrease in the rotor core resistance $R_c$, as noted in Table 3. On the other hand, an increase in the RMS value of the stator line current is due to an increase in the rotor current caused by the change in the value of rotor resistance referred to in the stator.

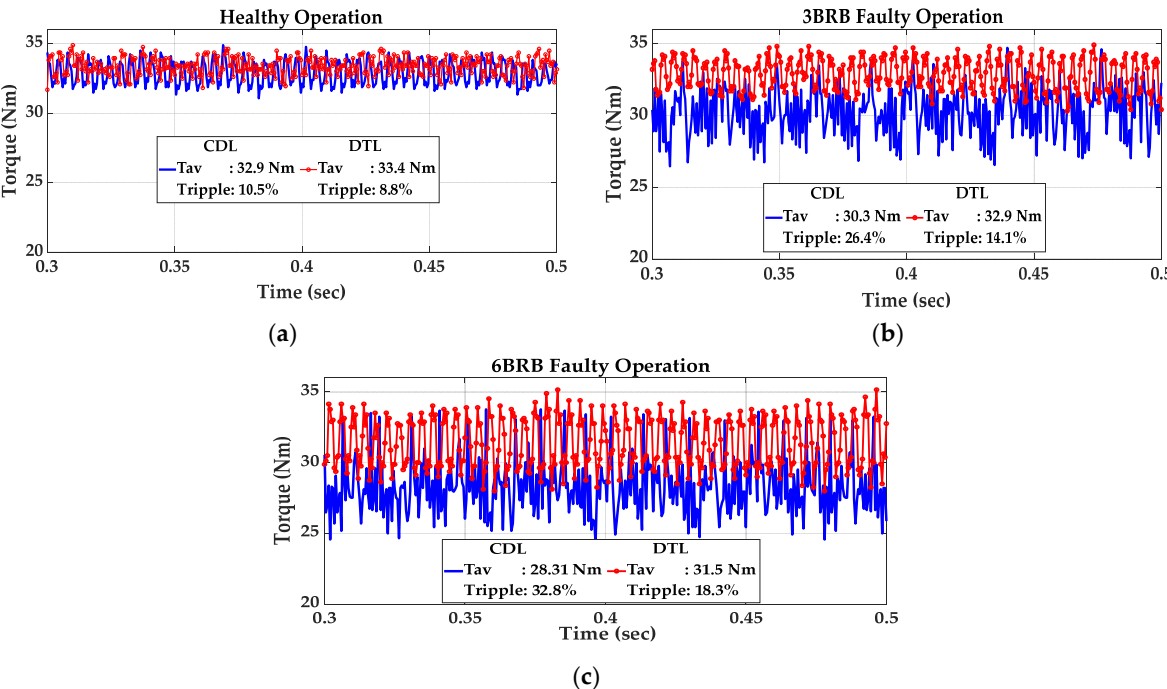

**Figure 22.** Shaft torque of loaded SCIMs: (**a**) healthy operation, (**b**) 3BRB faulty operation, (**c**) 6BRB faulty operation.

**Table 5.** Performance parameters of the SCIM with the CDL.

| Parameters | Healthy | | 3BRB | | 6BRB | |
|---|---|---|---|---|---|---|
| | **EXP** | **FEA** | **EXP** | **FEA** | **EXP** | **FEA** |
| $T_{av}(Nm)$ | 32.9 | 34.3 | 30.3 | 32.1 | 28.3 | 29.7 |
| $T_{ripple}$ (%) | 10.5 | 14.6 | 26.4 | 31.4 | 31.5 | 37.9 |
| $I_{steady}(A)$ | 9.60 | 8.64 | 10.40 | 10.14 | 11.11 | 11.43 |
| $I_{start}(A)$ | 84.8 | - | 78.8 | - | 76.4 | - |
| $I_{start}/I_{steady}$ | 8.83 | - | 7.57 | - | 6.87 | - |
| Eff (%) | 86.2 | 87.5 | 84.7 | 86.2 | 81.9 | 82.6 |
| $P_{TCu}(W)$ | 508.2 | 437.2 | 566.8 | 496.5 | 607.8 | 523.1 |
| $P_{Ti}(W)$ | 230.4 | 275.9 | 265.4 | 281.0 | 263.8 | 280.7 |
| $P_{add}(W)$ | 39.1 | - | 36.24 | - | 35.04 | - |
| PF (per unit) | 0.839 | 0.857 | 0.831 | 0.849 | 0.807 | 0.828 |

**Table 6.** Performance parameters of the SCIM with the DTL.

| Parameters | Healthy | | 3BRB | | 6BRB | |
|---|---|---|---|---|---|---|
| | **EXP** | **FEA** | **EXP** | **FEA** | **EXP** | **FEA** |
| $T_{av}(Nm)$ | 33.4 | 35.1 | 32.9 | 33.7 | 31.5 | 32.3 |
| $T_{ripple}(\%)$ | 8.8 | 12.9 | 14.1 | 19.21 | 18.3 | 23.7 |
| $I_{steady}(A)$ | 8.44 | 8.88 | 10.41 | 9.42 | 10.75 | 11.32 |
| $I_{start}(A)$ | 80.2 | - | 77.5 | - | 72.7 | - |
| $I_{start}/I_{steady}$ | 9.50 | - | 7.43 | - | 6.76 | - |
| Eff(%) | 86.9 | 88.1 | 85.1 | 87.2 | 82.3 | 85.7 |
| $P_{Tcu}(W)$ | 537.5 | 421.8 | 549.9 | 478.0 | 582.7 | 506.8 |
| $P_{Ti}(W)$ | 221.0 | 242.4 | 253.5 | 267.9 | 262.6 | 274.2 |
| $P_{add}(W)$ | 37.7 | - | 37.01 | - | 36.34 | - |
| PF(perunit) | 0.849 | 0.89 | 0.833 | 0.842 | 0.802 | 0.820 |

The rotor resistance, referred to in the stator, is also dependent on the slip. The latter depends on the frequency of the rotor current. An increase in stator and rotor currents caused the total copper losses to increase, thus decreasing the efficiency when operating with BRB faults. In all cases, the efficiencies of the SCIMs with a DTL winding configuration are slightly high compared with the efficiencies of the SCIMs with a CDL winding configuration. The SCIM with a DTL configuration has a low rotor resistance, as referred to in the stator, which has a slightly reduced line current when operating with BRB faults. Another observation to be made is the change in the total leakage reactance referred to in the stator. This parameter has an impact on the magnetic saturation level in the stator and rotor cores, thus affecting the total iron losses. The SCIM with a DTL winding configuration has reduced values of total leakage reactances referred to in the stator compared to the SCIM with a CDL winding configuration. This has led the SCIM with a DTL winding configuration to operate in all cases with reduced total iron losses, thus slightly increasing the efficiency. A decrease in the power factor is also notable when the motors operate with BRB faults. The SCIMs started directly online at the rated voltage and supply frequency. The RMS values of the FEA starting currents are not furnished because the initial currents in the AC magnetic transient models were kept at zero.

## 5. Conclusions

This article presents a performance analysis of a three-phase squirrel cage induction motor with a double–triple layer stator winding configuration operating with broken rotor bar faults. The squirrel cage induction motor with a double–triple layer stator winding configuration is modeled using a 2D finite element method. The results obtained from the 2D finite element method are validated through experimental results of a prototype tested under healthy and broken rotor bar faulty operations. The results show that the squirrel cage induction motor with a double–triple layer stator winding can mitigate some of the current components that have been induced in the stator winding when operating with broken rotor bar faults, especially the magnitude of sidebands for both unloaded and loaded currents. This is an indication that the SCIM with a DTL winding configuration can reduce the severity of broken rotor bar faults under unloaded and loaded conditions. The analysis in this article has proven that the stators of SCIMs destined for applications where broken rotor bar faults are frequent and where motors are required to continue operating until scheduled maintenance is in effect can be wound with double–triple layer winding configurations. Future work will include a thermal analysis of the stator winding and motor housing in a state of equilibrium and study the effect of broken rotor bars during the transient regime.

**Funding:** This research received no external funding.

**Data Availability Statement:** Data available on request due to restrictions e.g., privacy or ethical. The data presented in this study are available on request from the corresponding author.

**Conflicts of Interest:** The authors declare no conflict of interest.

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
