# Peer review of "Analysis of a Three-Phase Induction Motor with a Double–Triple-Layer Stator Winding Configuration Operating with Broken Rotor Bar Faults"

_machines, doi:10.3390/machines11111023_

Round 1
Reviewer 1 Report
Comments and Suggestions for Authors
The article discusses a failure tolerant design of an induction motor with a double-triple layer stator winding. It has been theoretically shown that for a motor with such a winding, in the event of significant damage to the rotor bars, the reduction in performance is significantly less than for a traditional motor with a two-layer winding. The paper also presents comprehensive experimental verification. The article may be of interest to researchers in the field of electrical machine design. Although, in my opinion, that the article should also cover the following questions:
1) Figure 12 shows that you examine cases of 3 broken bars and 6 broken bars, which are rare in practice. Is it possible to compare the characteristics of the CDL and DTL motors for the more common case of one broken bar?
2) Tables 5 and 6 show that the measured efficiency of the DTL motor is significantly higher than that of the CDL motor for all cases considered. It would be instructive to explain how the use of the double-triple layer stator winding can increase efficiency even in the case of a healthy motor. You should also add to these tables a comparison of individual types of losses: in copper, in steel, etc.
3) Broken rotor bars are also known to significantly increase overheating in induction motors. The article should also include the results of measuring the temperature of the stator winding and motor housing in a state of thermal equilibrium.
Author Response
Dear Reviewer,
I would like to thank you for the comments and recommendations. They have been very helpful as far as the strengthening of this manuscript is concerned. Herewith enclosed the responses.
Yours Sincerely,

Reviewer 2 Report
Comments and Suggestions for Authors
In this paper, the author investigates a three-phase Squirrel Cage Induction Motor (SCIM) with a Double-Triple Layer (DTL) stator winding configuration operating under Broken Rotor Bars (BRB) faults using the finite difference method to emulate potential failures. The overarching motivation for this research is of considerable interest and finds applications across various industrial sectors. To perform simulations via the finite difference method, the author employs the ANSYS 22.0 electromagnetic package, configuring realistic parameters in the simulation. The author obtains data on flux as a function of rotor angle and magnitude as a function of harmonic order for subsequent analysis. Additionally, experimental tests are conducted, and corresponding photographs are included in the manuscript. This approach effectively integrates numerical tools with experimental findings to advance the intriguing motivation of the study.
Nevertheless, prior to manuscript acceptance, the author should incorporate related works that have explored similar studies using alternative methodologies. This inclusion will elevate the standard of the research by connecting it more closely with the existing literature, while also enhancing the overall scholarly integrity of the literature review. The following references, along with their respective citations, serve as illustrative examples for our paper:
1) C. Pezzani, P. Donolo, G. Bossio, M. Donolo, A. Guzmán, and S. E. Zocholl, “Detecting broken rotor bars with zero-setting protection,” in *IEEE Transactions on Industry Applications*, vol. 50, no. 2, pp. 1373–1384, March 2014.
2) Bezerra de Deus, Danyelson Barros; Sobrinho, Carlos Alberto Nobrega; Belo, Francisco Antonio; Brito, Alisson V.; De Souza Ramos, Jorge Gabriel Gomes; Ramos, J. G.; Lima-Filho, Abel Cavalcante, "Density of Maxima Approach for Broken Bar Fault Diagnosis in Low Slip and Variable Load Conditions of Induction Motors," in *IEEE TRANSACTIONS ON INSTRUMENTATION AND MEASUREMENT*, vol. 69, pp. 9797-9804, 2020.
3) D. Morinigo-Sotelo, R. de J. Romero-Troncoso, P. A. Panagiotou, J. A. Antonino-Daviu, and K. N. Gyftakis, “Reliable detection of rotor bars breakage in induction motors via music and zsc,” in *IEEE Transactions on Industry Applications*, vol. 54, no. 2, pp. 1224–1234, March 2018.
4) J. Amezquita-Sanchez, M. Valtierra-Rodriguez, C. Pérez-Ramírez, D. Camarena-Martinez, A. Garcia-Perez, and R. Romero-Troncoso, “Fractal dimension and fuzzy logic systems for broken rotor bar detection in induction motors at start-up and steady-state regimes,” in *Measurement Science and Technology*, vol. 28, April 2017.
Once these minor adjustments have been incorporated, we recommend that the article be published in the journal "Machines."
Author Response

(The authors gave the same response as above.)

Reviewer 3 Report
Comments and Suggestions for Authors
The proposed work describes the Analysis of a Three-Phase Induction Motor with a Double-Triple 2-layer stator Winding Configuration Operating with Broken Ro-3 tor Bar Faults 4. Although the authors have covered all the aspects, the following modifications need to be considered for the possible publication of the manuscript;
1. The title needs to be checked grammatically. The prepositions are missing somewhere.
2. Line 8-10: (Abstract section) - It does not convey its intended meaning.
3. The abstract section should be completely restructured. It should consist of an introduction, objective, proposed methodology and conclusion with obtained results in a precise way.
4. The paper structure also needs to be modified. Section 1: Introduction, Section 2: Related Work, Section 3: Materials and Methodology (Including Component Specifications), Section 4: Results and Discussion, Section 5: Conclusion with Future Work.
5. Include the overall framework schematic sketch in the methodology section.
6. Overall work is good, but presentation should be improved.
Comments on the Quality of English Language
Extensive English editing is required including prepositions, active and passive voice, sentence structuring, etc.
Author Response

(The authors gave the same response as above.)
